# Exploring Information Exchange between *Thesium chinense* and Its Host *Prunella vulgaris* through Joint Transcriptomic and Metabolomic Analysis

**DOI:** 10.3390/plants13060804

**Published:** 2024-03-12

**Authors:** Anping Ding, Ruifeng Wang, Juan Liu, Wenna Meng, Yu Zhang, Guihong Chen, Gang Hu, Mingpu Tan, Zengxu Xiang

**Affiliations:** 1College of Horticulture, Nanjing Agricultural University, Nanjing 210095, China; 2021804301@stu.njau.edu.cn (A.D.); 2021804292@stu.njau.edu.cn (R.W.);; 2College of Life Sciences, Nanjing Agricultural University, Nanjing 210095, China

**Keywords:** *Thesium chinense*, *Prunella vulgaris*, transferred metabolite, mobile gene, haustoria formation

## Abstract

Background: *Thesium chinense* known as the “plant antibiotic” is a facultative root hemi-parasitic herb while *Prunella vulgaris* can serve as its host. However, the molecular mechanisms underlying the communication between *T. chinense* and its host remained largely unexplored. The aim of this study was to provide a comprehensive view of transferred metabolites and mobile mRNAs exchanged between *T. chinense* and *P. vulgaris*. Results: The wide-target metabolomic and transcriptomic analysis identified 5 transferred metabolites (ethylsalicylate, eriodictyol-7-O-glucoside, aromadendrin-7-O-glucoside, pruvuloside B, 2-ethylpyrazine) and 50 mobile genes between *T. chinense* and *P. vulgaris*, as well as haustoria formation related 56 metabolites and 44 genes. There were 4 metabolites (ethylsalicylate, eriodictyol-7-O-glucoside, aromadendrin-7-O-glucoside and pruvuloside B) that are transferred from *P. vulgaris* to *T. chinense*, whereas 2-ethylpyrazine was transferred in the opposite direction. Furthermore, we inferred a regulatory network potentially involved in haustoria formation, where three metabolites (N,N′-Dimethylarginine/SDMA, NG,NG-Dimethyl-L-arginine, 2-Acetoxymethyl-anthraquinone) showed significant positive correlations with the majority of haustoria formation-related genes. Conclusions: These results suggested that there was an extensive exchange of information with *P. vulgaris* including transferred metabolites and mobile mRNAs, which might facilitate the haustoria formation and parasition of *T. chinense*.

## 1. Introduction

Parasitic plants display remarkable diversity and are commonly categorized into two primary groups: holoparasites and hemiparasites, depending on their photosynthesis capabilities. Additionally, they are further distinguished as either root parasites or stem parasites based on the location of parasitism on the host plant [1,2]. Despite their remarkable diversity, all parasitic plants share a unique specialized organ called the haustorium [3], which has been described as “the essence of parasitism”. The haustorium plays a crucial role in the interaction between parasitic plants and their hosts. Early in the commensal process, it facilitates the parasite’s attachment and invasion of the host, and subsequently, it enables the uptake of nutrients, hormones and signaling molecules [4]. The symplastic continuity allows for the transfer of macromolecules and genetic materials between the hosts and the parasites [5].

*Thesium chinense*, a hemiparasitic plant within the genus *Thesium* of the Santalaceae family, exhibits a widespread distribution across Africa, Europe, Asia and America [6]. Contemporary pharmacological investigations have proved that *T. chinense* possesses anti-inflammation properties [7,8], antimicrobial effect [9,10,11], analgesic activity [12], antioxidant activity [13] and anti-nephropathy [14]. Termed as a “plant antibiotic” [15], *T. chinense* demonstrates therapeutic potential in addressing various conditions including mastitis, tonsillitis, pharyngitis, pneumonia, and upper respiratory tract infections [9,16,17]. *T. chinense* establishes parasitic associations with a diverse array of host plants [18], one of which is *Prunella vulgaris*, a perennial herb from the Lamiaceae family [19], by attaching itself to the roots through haustoria for sustenance and growth purposes.

Due to their unique symbiotic relationship, parasitic plants not merely absorb water [20] and nutrients from the host but also utilize secondary metabolites, mRNA [21], proteins [22], and systemic signals [23,24]. The haustorium functions as a vital conduit, facilitating bidirectionally exchange between the parasite plants and its hosts [25]. For example, *Cistanche deserticola* effectively utilizes metabolites derived from its host, *Haloxylon ammodendron*, to enhance its survival strategies [26]. *Cuscuta* not only transfers mRNAs and proteins between different host plants [27] but also exchanges proteins with similar functions among different host plants of *Cuscuta* [28]. Parasitic planta may actively manipulate host physiology by transferring phytohormones [23]. Recent research efforts on *T. chinense* have primarily concentrated on exploring its vitro anti-inflammatory and antimicrobial activity using extracts [7,29,30], host range and selectivity [18] and understanding the developmental reprogramming involved in haustoria formation [31]. Nevertheless, there remains a scarcity of studies examining the intricate information exchange between *T. chinense* and its host plants. Delving deeper into the molecular mechanisms governing this interaction is crucial for comprehending the successful parasitism and subsequent symbiosis between parasitic plants and their hosts.

To delve into the intricate information exchange events occurring between *T. chinense* and *P. vulgaris*, we conducted an integrated wide-target metabolomic and transcriptomic analysis. In this study, 5 transferred metabolites and 50 mobile genes were identified between *T. chinense* and *P. vulgaris*. Additionally, we discovered 56 metabolites and 44 genes that are intricately linked to haustoria formation. Thus, this study not only explores the information exchange events between *T. chinense* and its host, *P. vulgaris*, but also provides insights into haustoria formation and host invasion, shedding light on the intricate interplay between parasite and host during parasitism.

## 2. Results

### 2.1. Root Morphology of T. chinense and Its Host P. vulgaris Post Parasition

The root morphology of individual *T. chinense* and its host *P. vulgaris*, and the chimeric root post symbiosis were histologically observed (Figure 1A). The results revealed a significant presence of ivory spherical haustoria at the root of *T. chinense* (Figure 1B). Although the roots of *T. chinense* were tightly attached to the roots of *P. vulgaris*, the haustoria did not completely penetrate the roots of *P. vulgaris* (Figure 1C), implying that the bridge between *P. vulgaris* chimera and the haustorium was undergoing changes and transmitting cargos (Figure 1D). To explore the information exchange mechanisms between *T. chinense* and its host *P. vulgaris*, *T. chinense* chimera (THC) and *P. vulgaris* chimera (PC) from the symbiont roots, and the root counterparts of independent *T. chinense* (TH) and *P. vulgaris* (P) seedlings were collected for subsequent transcriptomic and metabolomic analysis.

### 2.2. Metabolomic Changes in T. chinense and Its Host P. vulgaris Post Symbiosis

To identify the metabolites transferred between *T. chinense* and its host *P. vulgaris*, the wide-target metabolomic analysis was conducted. Consequently, 1014 metabolites were identified in *T. chinense*, *P. vulgaris* and their chimeras (Figure 2A, Appendix A). Furthermore, a principal component analysis PCA of these metabolites demonstrated their clear segregation into four distinct clusters, corresponding to the four sampling groups (Figure 2B). These results suggested significantly different pattern of metabolites accumulation among TH, THC, PC, and P, emphasizing the profound impact of parasitism on metabolite profiles.

Subsequently, the differentially accumulated metabolites (DAMs) in *T. chinense* and its host *P. vulgaris* post symbiosis were identified using the screening criteria of |log2FoldChange| ≥ 1 and VIP ≥ 1. Compared with the roots of intact *T. chinense*, 252 DAMs were identified in *T. chinense* chimera, of which 75 upregulated and 177 downregulated metabolites (Appendix A). Similarly, a total of 194 DAMs were observed in *P. vulgaris* chimera compared to parasitism-free *P. vulgaris*, with 159 upregulated while 35 downregulated (Appendix A).

Regarding the DAMs category, phenolic acids, amino acids and derivatives, flavonoids and alkaloids collectively comprised a significant portion, exceeding half of the total DAMs detected in the TH vs. THC group. Among these, phenolic acids exhibited the highest percentage, accounting for 27.38% of the DAMs (Figure 3A). Notably, the majority of phenolic acids present in *T. chinense* chimera displayed a decreasing trend compared to *T. chinense*. However, a notable exception was ethylsalicylate, which exhibited higher accumulation in *T. chinense* chimera. Furthermore, a total of 23 flavonoids were identified, with the majority of DAMs associated with flavonoid biosynthesis, including kaempferol derivatives were downregulated in *T. chinense* chimera. Another notable issue is that most of the auxin biosynthesis related components, including indole, 3-indolepropionic acid, and 3-indoleacrylic acid were predominantly downregulated in *T. chinense* chimera (Appendix A).

In the P vs. PC comparison group, DAMs in the category of phenolic acids, flavonoids and terpenoids accounted for 14.43%, 5.15% and 7.73%, respectively (Figure 3B). Compared to *P. vulgaris*, ferulic acid methyl ester and p-coumaric acid methyl ester accumulated more in the *P. vulgaris* chimera, whereas the accumulation of protocatechuic acid, salicylic acid-2-O-glucoside, and arbutin was in the contrast trend. After symbiosis, the content of most flavonoids increased in *P. vulgaris* chimera. Interestingly, terpenoids (including kaurenoic acid, 18-oxoferruginol, and serratagenic acid) and jasmonic acid (JA) were all upregulated in *P. vulgaris* chimera (Appendix A).

Notably, 56 common DAMs were altered both in PC and THC compared to their respective uninfected roots. Therefore, these 56 DAMs could be regarded as haustoria formation related metabolites (Table 1, Figure 2C). In terms of haustoria formation related hormones among these 56 DAMs, there was a significant accumulation of auxin biosynthesis related components in PC, whereas the opposite was observed in THC. Jasmonic acid (JA) showed upregulation in both THC and PC. Moreover, another 16 metabolites were also synchronized upregulated in both chimeras’ groups, possibly promoting haustoria formation. Conversely, 11 metabolites were downregulated in both chimeras’ groups, suggesting they might inhibit haustoria formation (Table 1, Appendix A).

### 2.3. The Exchanges of Metabolites between T. chinense and Its Host P. vulgaris during Parasitism

To investigate the intricate information exchange between *T. chinense* and its host *P. vulgaris*, the accumulation pattern of metabolites in the four groups (TH, THC, PC, and P) were compared. Specifically, metabolites that were undetected in TH or P but were observed in other three samples were defined as transferred metabolites. Consequently, a total of 5 transferred metabolites (ethylsalicylate, eriodictyol-7-O-glucoside, aromadendrin-7-O-glucoside, pruvuloside B, 2-ethylpyrazine) were identified (Table 2). Notably, pruvuloside B, a characteristic component of *P. vulgaris*, was detected in PC, P and THC, however it was absent in TH roots, suggesting a transfer of this metabolite from *P. vulgaris* chimera to *T. chinense* chimera (host → parasite direction). Similarly, ethylsalicylate, eriodictyol-7-O-glucoside, and aromadendrin-7-O-glucoside were identified as host → parasite mobile metabolites. Conversely, 2-ethylpyrazine was presented in TH, THC, and PC but absent in P roots, indicating it as a metabolite transferred in the parasite to host direction.

### 2.4. Transcriptomic Changes in T. chinense and Its Host P. vulgaris Post Symbiosis

Besides the metabolomic fluctuation, the parasitism of *T. chinense* also induced significant transcriptomic changes. To systematically investigate these changes, transcriptomic profiling was performed on root samples from TH, THC, PC and P. The subsequent analysis was based on the Combined unigene dataset encompassing all these 4 samples. Then a stringent cutoff (|log2FoldChange| ≥ 1 with the adjusted *p*-value padj < 0.05) was used to identify differentially expressed genes (DEGs) in *T. chinense*, *P. vulgaris* and their chimeras post parasition. Consequently, 11,640 and 8705 DEGs were identified in the comparison of TH vs. THC (Appendix A) and P vs. PC (Appendix A), respectively.

To infer the biological functions of DEGs of *T. chinense* and its host *P. vulgaris* post symbiosis, the GO and KEGG enrichment analysis of DEGs were performed. Regarding the DEGs in TH vs. THC group, the GO entries and proportions with the most significant enrichment in biological process, cellular component, and molecular function were photosynthesis/light reaction, photosystem and hydrolase activity/hydrolyzing N-glycosyl compounds, respectively (Figure 4A), while the three most significant counterparts in P vs. PC group were amino acid transport, ER body, and organic acid binding (Figure 4B). The KEGG enriched pathways of DEGs in TH vs. THC and P vs. PC were similar, both including phenylpropanoid biosynthesis, flavonoid biosynthesis and plant hormone signal transduction. In addition, the DEGs in the TH vs. THC group were also highly enriched in fructose and mannose metabolism, vitamin B6 metabolism and photosynthesis-antenna proteins (Figure 4C). However, the highly represented pathways of DEGs in P vs. PC were plant-pathogen interaction and MAPK signaling pathway-plant (Figure 4D).

### 2.5. The Mobile Genes between T. chinense and Its Host P. vulgaris

To delve deeper into the molecular-level information exchange events between *T. chinense* chimera and its host *P. vulgaris* chimera, we performed a stepwise bioinformatic classification to identify mobile transcripts between parasite plant and its host. The Combined unigene dataset were filtered with BLAST against the genome sequence of *Santalum yasi* and *P. vulgaris*. Consequently, 9411 genes were finally retrieved from *Santalum* and 9814 genes from *P. vulgaris* (Appendix A).

To accurately discern the origin of these genes, we employed the criteria that genes with FPKM < 3 in the P but FPKM ≥ 3 in other three groups (TH, THC, PC) were considered as being originated from *T. chinense*. As a result, 44 genes were identified as mobile transcripts transferred from *T. chinense* to *P. vulgaris* chimera, denoted as Th → P mobile genes. Likewise, 6 genes were mobile genes transferred in the opposite direction from *P. vulgaris* to *T. chinense* (P → Th) (Table 3).

### 2.6. The Conjoint Analysis of Genes and Metabolites Related to Haustoria Formation

To identifying genes closely related to haustoria formation, unigenes in the intersection of THC and PC were retrieved from the Combined unigene dataset encompassing all 4 samples through filtering the BLAST results, and 189 common genes were obtained (Appendix A).

To systematically understand the metabolite-gene relationships ascribed to haustoria formation, we constructed the metabolite-gene network map with the threshold of |coefficient| > 0.8. Out of the 189 genes in the intersection of THC and PC, 44 genes were selected using the criteria of upregulated expression in both chimera (Table 4) to analyze their correlation with 56 metabolites related to haustoria formation. Subsequently, this narrowed down the search to 21 genes and 26 metabolites for constructing the correlation network map (Appendix A).

Further analysis of the gene-metabolite correlation network related to haustoria formation showed that three metabolites (N,N′-Dimethylarginine/SDMA, NG,NG-Dimethyl-L-arginine, 2-Acetoxymethyl-anthraquinone) were significantly positively correlated with the majority of haustoria formation-related genes, while 2,2-Dimethylsuccinic acid was only positively correlated with only one gene (*ACT1_ORYS)*. These positive correlation of genes and metabolites may synergistically participate in the formation of haustoria during the parasition process of *T. chinense*, helping it successfully parasitize *P. vulgaris* (Figure 5).

## 3. Discussion

*T. chinense* is a medically important plant that invades its host through the haustoria and hijacks water, nutrients, DNA, mRNA, proteins needed to sustain its own growth and development. The essence of parasite plants’ life habits is to establish parasitic relationships with their host [32]. However, there have been few studies on the information exchange between *T. chinense* and its host thus far. Therefore, this study aims to explore the changes in metabolome and transcriptome of both *T. chinense* and its host *P. vulgaris*, as well as the transferred of metabolites and mobile genes between them.

According to the current phytochemical investigations available, *T. chinense* contains various a diverse range of compounds, with flavonoids being the main biologically active compounds responsible for its pharmacological properties and therapeutic efficacy [7]. In this study, it was observed that *T. chinense* chimera showed a higher proportion of downregulated flavonoids compared to individual *T. chinense* (Appendix A). This could be a result of plant growth-defense trade-off where part of plant resources, originally allocated to growth, were redirected towards defense mechanisms, thus obtaining protective adaptation to environmental stresses. The bioactive compounds of *P. vulgaris* predominantly comprise flavonoids, phenolic acids, and terpenoids [33]. After establishing a parasitic relationship, most flavonoids and terpenoids showed an upregulation trend (Appendix A). This result indicates that parasitism promotes the accumulation of active compounds in *P. vulgaris*. These results provide a basis for understanding the metabolic mechanisms of *T. chinense*-*P. vulgaris* interactions, which will contribute to the quality control of *T. chinense*.

Phytohormones play a crucial role in regulating plant growth and development [23]. By analyzing the KEGG pathway, many DEGs in TH vs. THC and P vs. PC were found to be enriched in plant hormone signal transduction (Figure 4C,D). Recent studies have showed the formation of plant hormones such as auxin, cytokinins, and ethylene in haustorium formation [34]. Once invasion is successfully, haustoria start the formation of xylem bridges to facilitate material transfer between host and parasite xylems. This process is supported by auxin flow generated by several PIN family auxin efflux carriers and AUX1/LAX influx carriers genes expressed within invading haustoria [35]. Haustorium-inducing factors (HIFs) trigger the expression of an auxin biosynthesis gene in root epidermal cells at the sites where haustoria formation occurs. This process leads to cell division and expansion, resulting in the formation of a semi-spherical pre- or early haustorium structure [31]. Therefore, there is a high abundance of auxin biosynthesis/signaling-related genes in *T. chinense* haustoria [36]. Furthermore, it is plausible that the involvement of auxin response serves as a shared mechanisms for haustoria formation among parasitic plants [37]. In the present study, the levels of auxins such as indole, 3-indolepropionic acid and 3-indoleacrylic acid decreased in *T. chinense* chimera (Table 1). Similarly, during its parasitization process, *Cuscuta japonica* also exhibited a decline in auxin content [23]. The auxin pathway may play an important role in the association host and parasite [37]. Therefore, auxin transport may participate in establishing the host-parasite association. JA, an ancient regulator controlling systemic signals biosynthesis and/or transport, plays a crucial role in the biosynthesis or transport of mobile signals between-plants [23]. Furthermore, the host JA signaling plays a role in regulating the gene expression in the parasitizing *Cuscuta* [37]. In this study, JA was upregulated in THC and PC (Table 1). However, the specific functions of JA in parasitic plants remain unexplored. We speculate that the increased level of JA in *T. chinense* chimera may be related to its defense mechanism against the host since the chimera also accumulating more JA. In short, the progression of haustorium organogenesis and the host-parasite interaction is controlled by phytohormones. To understand how plants coordinate multiple hormonal components in response to diverse developmental and environmental cues represents a significant challenge for the future. In our study, the metabolic changes caused by *T. chinense* parasitism were associated with phenylpropanoid biosynthesis, flavonoid biosynthesis, plant hormone signal transduction, fructose and mannose metabolism, vitamin B6 metabolism and photosynthesis-antenna proteins (Figure 4C). The fructose and mannose metabolism pathway is crucial for the success of parasitism [26]. In the case of *Orobanche aegyptiaca*, the host-induced suppression of the mannose 6-phosphate reductase gene is concomitant with significant mannitol decrease and increased tubercle mortality [38]. In plant-pathogen interaction, the pathogen secretes mannitol as a buffer against oxidative stress, and the host plant activates mannitol dehydrogenase to counter it [39]. In the study, the relatively high mannitol level in *P. vulgaris* chimera might be a consequence of this host-parasite interaction (Appendix A).

Parasitic plants and their hosts are often phylogenetically very distant, and the haustoria establish physical and physiological connections between the host and parasitic plants, thereby dominating most of their interactions [5], making the host-parasite systems very suitable for the identification of mobile substances. Secondary metabolites are essential for plant survival and are typically biosynthesized in specific tissues and cell types before being transported to neighboring cells or even to other tissues or other organs. Some secondary metabolites in the host can be transferred to the parasite plant [23]. We have identified 4 metabolites that were transferred from *P. vulgaris* chimera to *T. chinense* chimera (Table 2). In this study, 2-ethylpyrazine was identified to be the transferred metabolite from *T. chinense* chimera to *P. vulgaris* chimera (Table 2). Although what effect 2-ethylpyrazine has on the parasitism relationship remains unknown, we speculate that it may be a metabolite of *T. chinense* that attracts host plants and successfully colonizes them. Actually, how parasitic plants accept secondary metabolites from their hosts and the ecological impact of the translocated secondary metabolites in parasitic plants require further exploration [23]. In future experiments, we can apply 2-ethylpyrazine to *P. vulgaris* or other host plants of *T. chinense* and observe whether *T. chinense* can colonize faster or promote its growth to verify the role of 2-ethylpyrazine in contributing to establish parasitism relationship.

Compared to other host-pathogen systems [40], there have been relatively few reports on the interactions between parasite and plant-hosts [41]. An important aspect of this interaction is the influence of the host’s growth stage and environment on the expression of mobile mRNAs [21]. The presence of haustoria also facilitates the transfer of RNAs between parasitic plants and their hosts [42]. RNA-sequencing analysis has indicated the trafficking of thousands of mRNA species between hosts and *Cuscuta pentagona* [27]. Similarly, there has been significant mobile mRNA transfer between *Haloxylon ammodendron* and the parasitic plant *Cistanche deserticola* [41], with mRNA abundance likely playing a key role in determining mobility [23]. In this study, cross-species mRNA movement was identified between *T. chinense* and *P. vulgaris*, with 44 and 6 mobile mRNAs potentially being transferred from *T. chinense* and *P. vulgaris* to their respective hosts and parasite through haustoria (Table 3). Nonetheless, it remains to investigate whether these mobile mRNAs exert functional implications in host-parasite interactions. The elucidation of the underlying mechanisms governing the exchange of informational cues between plants remains an ongoing biological enigma.

## 4. Materials and Methods

### 4.1. Plant Materials and Sample Collection

Five plants each of *T. chinense* (TH), *P. vulgaris* (P) and their commensal chimera were randomly selected for sampling independent roots and chimeric roots, and the *T. chinense* chimera (THC) and *P. vulgaris* chimera (PC) were sampled from the symbiont roots post parasitization. To minimize any surface tissue contamination, the sampled roots or chimera with three biological replicates were washed 1–2 times with PBS/RNase-free water, and frozen in liquid nitrogen and stored at −80 °C for the subsequent metabolomic, transcriptomic analysis.

### 4.2. Metabolomic Analysis

For the widely targeted metabolomic profiling, four type of root samples aforementioned were freeze-dried by vacuum freeze-dryer (Scientz-100F, Ningbo, China). The freeze-dried sample was crushed using a mixer mill (MM 400, Retsch, Shanghai, China) with a zirconia bead for 1.5 min at 30 Hz. Dissolve 50 mg of lyophilized powder with 1.2 mL 70% methanol solution, vortex 30 s every 30 min for 6 times in total. Following centrifugation at 12,000 rpm for 3 min, the extracts were filtrated (SCAA-104, 0.22 μm pore size; ANPEL, Shanghai, China), then analyzed using an UPLC-ESI-MS/MS system (UPLC, ExionLC™ AD Framingham, MA, USA; MS, Applied Biosystems 6500 QTRAP, Foster, CA, USA) [43,44].

Based on the mass spectrometry data, metabolites were identified using the Metware Database (MWDB, Wuhan, China) (www.metware.cn, accessed on 20 October 2023) and quantified according to peak intensity. Both unsupervised principal component analysis (PCA) and orthogonal projections to latent structure-discriminant analysis (OPLS-DA) were used to observe the overall differences in metabolic profiles between groups to identify their significant differential metabolites. The quantification data of metabolites were normalized by unit variance scaling and used for the subsequent analysis (http://www.r-project.org, accessed on 20 October 2023) [45].

### 4.3. Screening of Differentially Accumulated Metabolites

To determine the metabolomic differences of *T. chinense* and its host post parasitization, the differentially accumulated metabolites (DAMs) in the TH vs. THC and P vs. PC groups were screened. Variable importance in projection (VIP) values were extracted from OPLS-DA results, those selected and metabolites with VIP ≥ 1 and absolute |log2FoldChange| ≥ 1 were defined as DAMs [46,47].

The DAMs were annotated using the KEGG Compound database (https://www.kegg.jp/kegg/compound, accessed on 25 October 2023) and mapped to the KEGG Pathway database (https://www.kegg.jp/kegg/pathway.html, accessed on 25 October 2023) [48]. Then a KEGG pathway enrichment analysis was performed, and the significance was determined by hypergeometric test *p*-values ≤ 0.05 [49].

### 4.4. RNA Extraction, Library Construction and Sequencing

Total RNA was isolated using the Trizol Reagent [50] (Invitrogen Life Technologies, Shanghai, China). To ensure the RNA samples were integrated and DNA-free, agarose gelelectrophoresis was performed. RNA purity was then determined by a nanophotometer. Following that, a Qubit 2.0 Fluorometer and an Agilent 2100 BioAnalyzer (Agilent Technologies, Palo Alto, CA, USA) were used to accurately measure RNA concentration and integrity, respectively. The qualified samples were processed with oligo (dT) beads to enrich the mRNA, which was broken into fragments and used as templates for the cDNA library. To qualify the cDNA library, the fluorometer was used for primary quantification and the bioanalyzer was then used to insert text size. The qualified library was sequenced using the Illumina HiSeq 6000 platform (San Diego, CA, USA).

### 4.5. RNA-Seq Analysis

Clean reads were obtained by eliminating low-quality reads and assembled using Trinity 2.8.5 software [51]. The transcripts were assembled and then clustered into unigenes, and 5 unigene datasets (TH, THC, P, PC and Combined) were obtained through 5 assembling processes. The method of fragments per kilobase of transcript per million fragments mapped (FPKM) was applied to calculate the expression levels of genes. DESeq2 was used to identify differential expression genes (DEGs) based on the thresholds of the adjusted *p*-value padj  < 0.05 and |log2FoldChange| ≥ 1 [52]. Then DEGs were annotated by the NR, SwissProt, GO, KOG, Pfam, and KEGG databases [53,54]. Finally, GO and KEGG pathway enrichment analysis were performed on DEGs to reveal functional modules and signal pathways of interest.

### 4.6. Integrated Metabolomic and Transcriptomic Analysis

Through comparing the accumulation of metabolites in the four groups (TH, THC, PC, and P), metabolites that were not detected in TH or P but accumulated in the other three samples were categorized as transferred metabolites according to the method of identifying mobile genes described previously [27,55].

The Combined unigene dataset were filtered with BLAST against *Santalum yasi* genome sequence (https://ngdc.cncb.ac.cn/gwh/Assembly/37825, 31 December 2023) [56] and *Prunella vulgari* genome sequence (https://www.ncbi.nlm.nih.gov/datasets/genome/GCA_026898435.1, 31 December 2023) with the threshold of E-value = 1 × 10^−10^ suggested in previous reports [27,41]. Then additional criteria of FPKM < 3 in intact sample (TH, P) but FPKM > 3 in other three samples were used to filter mobile genes.

Genes that were not detected in TH but present in the other three samples were classified as host → parasite mobile genes from *P. vulgaris* to *T. chinense* using the model described previously [27,41,55]. The parasite → host mobile RNAs only not detected in P samples were identified in a similar manner.

### 4.7. Constructing the Network of Metabolites and Genes Related to Haustoria Formation

The unigene dataset of TH, THC, PC, and P was compared to the Combined unigene dataset encompassing all the 4 samples using BLASTP with the threshold of E-value = 1 × 10^−10^ suggested in previous reports [27,41], and the filtered unigenes of TH, THC, PC, and P were subjected for the subsequent downstream analysis. Venn diagram analysis of unigenes in TH, THC, PC and P datasets was performed to identify common genes between THC and PC. Those genes upregulated in both chimera and FPKM < 0.3 [41] in intact sample (TH, P) were considered as relating to haustoria formation.

Utilizing metabolite content and gene expression data, Pearson correlation tests were employed to identify connections between genes and metabolites related to haustoria formation. Correlations between DAMs and DEGs were refined based on Pearson correlation coefficient (PCC) and *p*-value criteria. Only significant associations with |PCC|  >  0.80 and *p*-value  <  0.05 were selected for constructing network of metabolome and transcriptome. The metabolite-gene relationships related to haustoria formation were visualized using Cytoscape (v3.9.0) [57].

## 5. Conclusions

Our study provides a deep dive into the metabolome and transcriptome of *T. chinense*, *P. vulgaris* and their chimeras, shedding light on the intricate dynamics of their parasitic relationship. The identification of 5 transferred metabolites and 50 mobile genes exchanged between the two species highlights the extensive inter-organismal transfer of resources and genetic information, underscoring the complexity of their interaction. Moreover, the discovery of 56 metabolites and 44 genes associated with haustoria formation reveals the sophisticated biological processes involved in establishing parasitism. The regulatory network has revealed three metabolites were significantly positively correlated with the majority of haustoria formation-related genes, offering valuable insights into potential targets for further research on parasitic plant development mechanisms. Notably, our findings emphasize the critical role of the fructose and mannose metabolism pathway in the success of parasitism, indicating a strategic utilization of host resources essential for the survival and proliferation of parasitic plants.

In conclusion, our results suggest that *T. chinense* engages in a dynamic and intricate biological exchange with *P. vulgaris*, leveraging both metabolites and mobile mRNAs to drive haustoria formation and ensure successful parasitism. By unraveling these complex interactions, our study not only advances our understanding of the molecular dialogues between parasitic and host plants but also paves the way for future investigations aimed at manipulating or harnessing these interactions for agricultural and ecological benefits.

## Figures and Tables

**Figure 1 plants-13-00804-f001:**
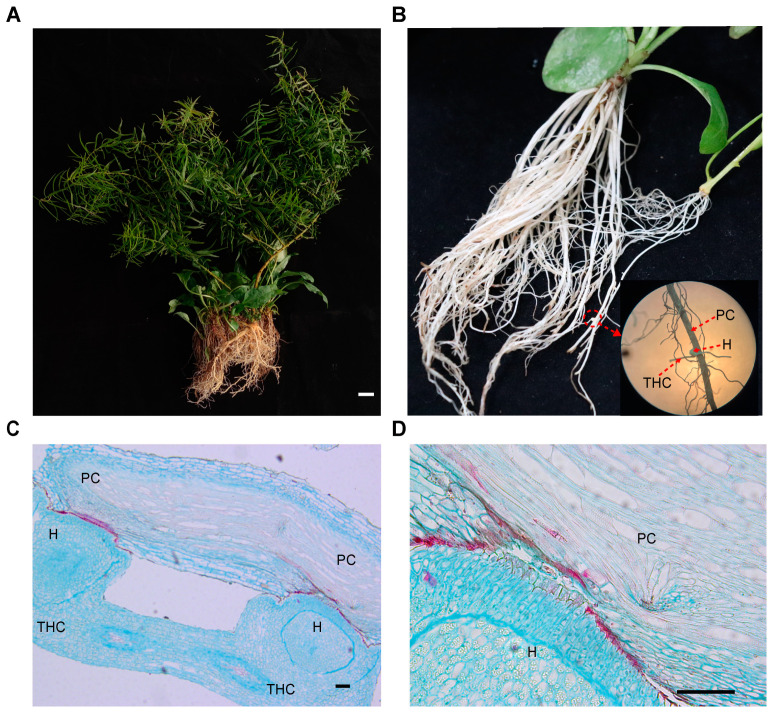
Morphology of *T. chinense* and its host *P. vulgaris* chimeric root. (**A**): *T. chinense* and its host *P. vulgaris* (scale bare: 1 cm). (**B**): *T. chinense* chimera is connected to its host *P. vulgaris* chimera through the haustorium. (**C**,**D**): Structure of *T. chinense* chimera, haustoria and *P. vulgaris* chimera (scale bare: 100 μm). THC: *T. chinense* chimera. H: Haustorium. PC: *P. vulgaris* chimera.

**Figure 2 plants-13-00804-f002:**
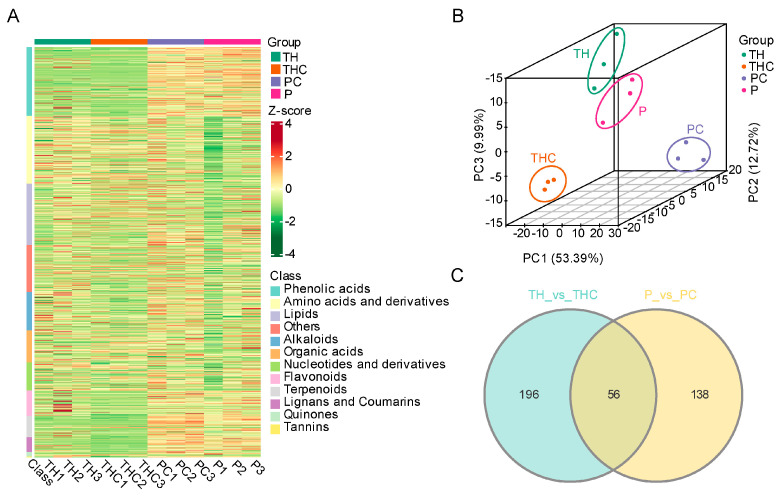
The metabolomic analysis of *T. chinense* and its host *P. vulgaris* post symbiosis. (**A**): Heat map visualization of metabolites in *T. chinense*, *P. vulgaris* roots and their chimera. (**B**): PCA analysis of metabolites in *T. chinense*, *P. vulgaris* roots and their chimera. (**C**): Venn diagrams revealing the relationship of differentially accumulated metabolites (DAMs) in *T. chinense* chimera and its host *P. vulgaris* chimera. TH: *T. chinense*. THC: *T. chinense* chimera. PC: *P. vulgaris* chimera. P: *P. vulgaris*.

**Figure 3 plants-13-00804-f003:**
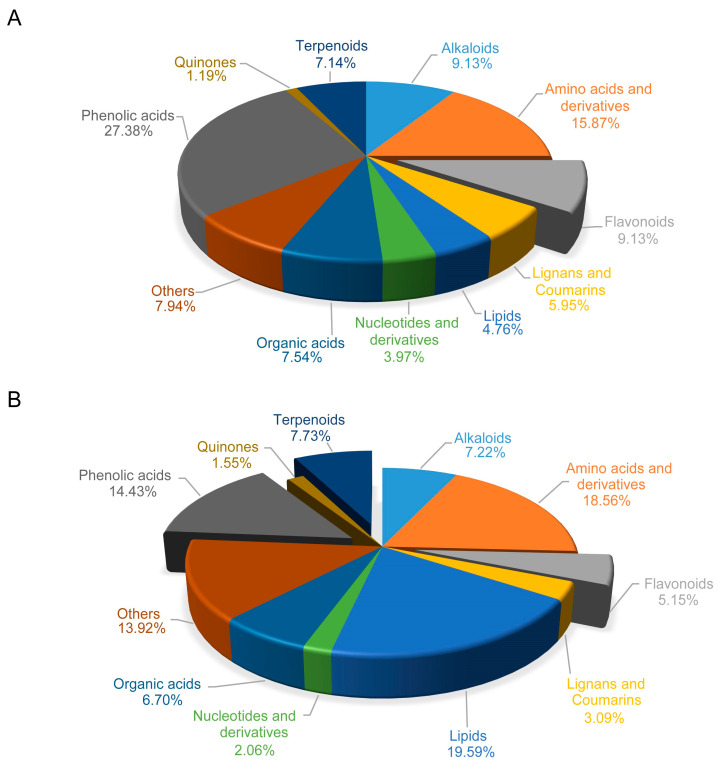
The proportion of differentially accumulated metabolites (DAMs) category in *T. chinense* chimera and its host *P. vulgaris* chimera. (**A**): TH vs. THC. (**B**): P vs. PC. TH: *T. chinense*. THC: *T. chinense* chimera. PC: *P. vulgaris* chimera. P: *P. vulgaris*.

**Figure 4 plants-13-00804-f004:**
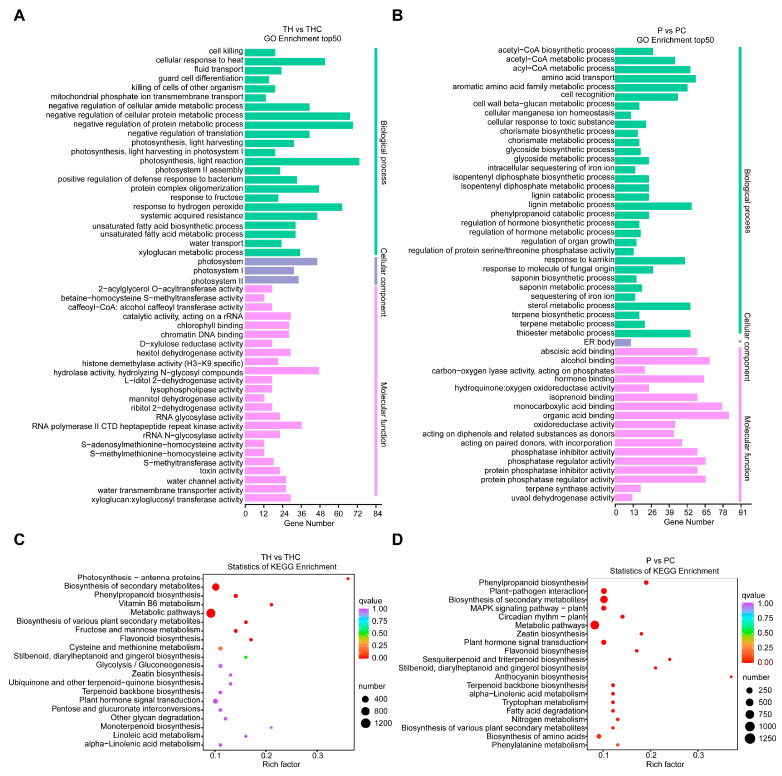
The enrichment analysis of DEGs based on GO terms and KEGG pathways. (**A**): GO terms of DEGs in TH vs. THC. (**B**): GO terms of DEGs in P vs. PC. (**C**): KEGG pathway analysis of DEGs in TH vs. THC. (**D**): KEGG pathway analysis of DEGs in P vs. PC. TH: *T. chinense*. THC: *T. chinense* chimera. PC: *P. vulgaris* chimera. P: *P. vulgaris*.

**Figure 5 plants-13-00804-f005:**
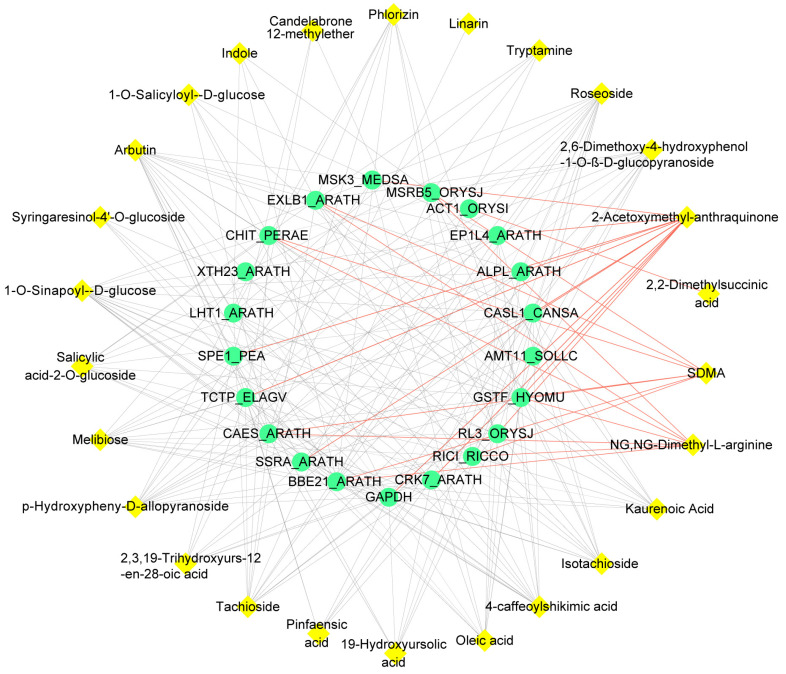
The correlation network of metabolites and genes related to haustoria formation. Metabolite and gene networks associated with haustorium formation. Green circles represent genes. Yellow diamonds represent metabolites. For associations between genes and metabolites, red lines represent positive correlations and gray lines represent negative correlations. The thickness of the line represented the correlation degree and the thicker the line, the higher the correlation. The correlation of haustoria formation related metabolites and genes are given in Appendix A.

**Table 1 plants-13-00804-t001:** Differentially accumulated metabolites (DAMs) related to haustoria formation.

Compounds	CAS	Category	Type	
			THvsTHC	PvsPC
2,3,19-Trihydroxyurs-12-en-28-oic acid	-	Terpenoids	up	up
Pinfaensic acid	-	Terpenoids	up	up
N,N′-Dimethylarginine; SDMA	30344-00-4	Amino acids and derivatives	up	up
N-Monomethyl-L-arginine	17035-90-4	Amino acids and derivatives	up	up
L-Isoleucyl-L-Aspartate	-	Amino acids and derivatives	up	up
L-Aspartyl-L-Phenylalanine	13433-09-5	Amino acids and derivatives	up	up
Candelabrone 12-methyl ether	-	Terpenoids	up	up
19-Hydroxyursolic acid	-	Terpenoids	up	up
Homoarginine	156-86-5	Amino acids and derivatives	up	up
NG,NG-Dimethyl-L-arginine	30315-93-6	Amino acids and derivatives	up	up
2-Deoxyribose-1-phosphate	17210-42-3	Nucleotides and derivatives	up	up
6′-O-Feruloyl-D-sucrose	118230-77-6	Phenolic acids	up	up
Jasmonic acid	77026-92-7	Organic acids	up	up
2-Acetoxymethyl-anthraquinone	-	Quinones	up	up
Propyl 4-hydroxybenzoate	94-13-3	Phenolic acids	up	up
5-hydroxy-1-phenyl-7-3-heptanone	-	Others	up	up
2,2-Dimethylsuccinic acid	597-43-3	Organic acids	up	up
L-Tartaric acid	87-69-4	Organic acids	up	down
Tachioside	109194-60-7	Phenolic acids	down	down
Isotachioside	31427-08-4	Phenolic acids	down	down
1-O-Salicyloyl-β-D-glucose	60517-74-0	Phenolic acids	down	down
Salicylic acid-2-O-glucoside	10366-91-3	Phenolic acids	down	down
p-Hydroxypheny-β-D-allopyranoside	-	Phenolic acids	down	down
Arbutin	497-76-7	Phenolic acids	down	down
Sinapoyl malate	92344-58-6	Phenolic acids	down	down
2-O-Caffeoylglucaric Acid	-	Phenolic acids	down	down
Oleic acid	112-80-1	Lipids	down	down
N-Methyl-Trans-4-Hydroxy-L-Proline	4252-82-8	Amino acids and derivatives	down	down
2,6-Dimethoxy-4-hydroxyphenol-1-O-ß-D-glucopyranoside	-	Others	down	down
Methoxyindoleacetic acid	3471-31-6	Alkaloids	down	up
Tryptamine	61-54-1	Alkaloids	down	up
L-Tryptophan	73-22-3	Amino acids and derivatives	down	up
3-Indoleacetonitrile	771-51-7	Alkaloids	down	up
1-Methoxy-indole-3-acetamide	-	Alkaloids	down	up
Indole	120-72-9	Alkaloids	down	up
3-Indolepropionic acid	830-96-6	Alkaloids	down	up
3-Indoleacrylic acid	1204-06-4	Alkaloids	down	up
γ-glutamylmethionine	17663-87-5	Amino acids and derivatives	down	up
2-Aminoethanesulfonic acid	107-35-7	Organic acids	down	up
p-Coumaric acid methyl ester	19367-38-5	Phenolic acids	down	up
Roseoside	54835-70-0	Others	down	up
Isoquinoline	119-65-3	Alkaloids	down	up
4-caffeoylshikimic acid	-	Phenolic acids	down	up
L-Histidine	71-00-1	Amino acids and derivatives	down	up
Phlorizin	60-81-1	Flavonoids	down	up
3,4-Methylenedioxy cinnamyl alcohol	58095-76-4	Lignans and Coumarins	down	up
Kaurenoic Acid	6730-83-2	Terpenoids	down	up
LysoPC 15:0	108273-89-8	Lipids	down	up
Melibiose	585-99-9	Others	down	up
3-amino-2-naphthoic acid	-	Alkaloids	down	up
L-Lysine-Butanoic Acid	80407-71-2	Amino acids and derivatives	down	up
cyclo-(Gly-Phe)	10125-07-2	Amino acids and derivatives	down	up
Trans-Citridic acid	4023-65-8	Organic acids	down	up
1-O-Sinapoyl-β-D-glucose	-	Phenolic acids	down	up
Linarin	480-36-4	Flavonoids	down	up
Syringaresinol-4′-O-glucoside	7374-79-0	Lignans and Coumarins	down	up

CAS: Chemical Abstracts Service registry number. TH: *T. chinense*. THC: *T. chinense* chimera. PC: *P. vulgaris* chimera. P: *P. vulgaris*.

**Table 2 plants-13-00804-t002:** The transferred metabolites between *T. chinense* and its host *P. vulgaris*.

Compounds	CAS	Category	TH	THC	PC	*p*
Ethylsalicylate	118-61-6	Phenolic acids	-	35,397	500,949	542,212
Eriodictyol-7-O-glucoside	38965-51-4	Flavonoids	-	29,411	3,851,867	4,694,065
Aromadendrin-7-O-glucoside	28189-90-4	Flavonoids	-	113,418	1,191,540	604,233
Pruvuloside B	-	Terpenoids	-	1,789	54,923	40,759
2-Ethylpyrazine	13925-00-3	Alkaloids	49,686	43,058	78,849	-

CAS: Chemical Abstracts Service registry number. TH: *T. chinense*. THC: *T. chinense* chimera. PC: *P. vulgaris* chimera. P: *P. vulgaris*.

**Table 3 plants-13-00804-t003:** The mobile genes between *T. chinense* and *P. vulgaris*.

Unigene ID	Gene	Annotation	FPKM			
			TH	THC	PC	*p*
***T. chinense* → *P. vulgaris***						
Cluster-12122.6	*R10A*	60S ribosomal protein L10a	924.6	1404.5	3.6	0.0
Cluster-13284.0	*RAN1*	GTP-binding nuclear protein Ran1	452.4	907.1	8.0	0.0
Cluster-14148.4	*TIC32*	Short-chain dehydrogenase TIC 32, chloroplastic	982.1	2251.9	4.4	0.0
Cluster-16015.3	*METK5*	S-adenosylmethionine synthase 5	433.0	3149.3	5.5	0.0
Cluster-16686.0	*WRK40*	Probable WRKY transcription factor 40	1142.9	1043.8	10.9	0.0
Cluster-16839.1	*SUNN*	Leucine-rich repeat receptor-like kinase	49.3	46.8	4.9	0.0
Cluster-1723.0	*UNC13*	Protein unc-13 homolog	32.9	46.3	6.2	0.0
Cluster-18022.0	*ORM1*	ATORM1, OROSOMUCOID-LIKE 1 ORM1	125.2	148.4	6.4	0.0
Cluster-22492.4	*PPA29*	Probable inactive purple acid phosphatase 29	917.8	1041.0	4.6	0.0
Cluster-23922.0	*LTI6B*	Hydrophobic protein LTI6B	365.2	891.6	4.6	0.0
Cluster-23995.0	*EMB8*	Embryogenesis-associated protein EMB8	88.3	49.9	11.0	0.0
Cluster-24355.1	*KNAP3*	Homeobox protein knotted-1-like 3	257.7	238.1	5.9	0.1
Cluster-24679.1	*BAGP1*	BAG-associated GRAM protein 1	37.1	74.0	3.7	0.0
Cluster-24856.0	*C7A12*	Cytochrome P450 CYP736A12	174.8	199.3	3.2	0.0
Cluster-25215.2	*EP1L3*	EP1-like glycoprotein 3	395.6	467.9	3.1	0.0
Cluster-25707.0	*PMT1*	Probable methyltransferase PMT1	28.0	67.0	5.1	0.0
Cluster-26296.0	*RTNLB*	Reticulon-like protein B2	226.7	269.3	9.2	0.1
Cluster-26397.0	*RL24*	60S ribosomal protein L24	56.8	102.6	3.8	0.0
Cluster-2641.0	*VP371*	Vacuolar protein-sorting-associated protein 37	104.5	151.3	11.6	0.0
Cluster-27486.3	*KAD7*	Probable adenylate kinase 7, mitochondrial	538.7	480.3	3.6	0.1
Cluster-28157.0	*DG*	DNA glycosylase superfamily protein	7.6	92.6	9.6	0.0
Cluster-28431.7	*ALFC6*	Fructose-bisphosphate aldolase 6, cytosolic	706.9	1620.3	3.6	0.0
Cluster-28487.3	*ADS3*	Palmitoyl-monogalactosyldiacylglycerol delta-7 desaturase	1432.6	1611.5	3.3	0.0
Cluster-30032.0	*GRP*	Glycine-rich protein A3	2030.6	2754.2	4.5	0.0
Cluster-31395.0	*TRAPPC3*	Transport protein particle (TRAPP)	95.3	98.7	10.1	0.0
Cluster-31481.0	*CAMT*	Caffeoyl-CoA O-methyltransferase	468.8	1485.6	5.5	0.0
Cluster-34709.1	*IF5A*	Eukaryotic translation initiation factor 5A	1030.8	908.8	3.1	0.0
Cluster-36607.0	*PRU1*	Major allergen Pru ar 1	2404.5	7930.2	8.7	0.3
Cluster-37697.0	*TCTP*	Translationally-controlled tumor protein	8294.9	9520.8	5.7	0.1
Cluster-4150.5	*NIN1*	Neutral/alkaline invertase 1, mitochondrial	195.7	202.0	3.8	0.0
Cluster-4583.0	*PER52*	Peroxidase 52	39.1	207.4	4.7	0.0
Cluster-6216.6	*GBLP*	Guanine nucleotide-binding protein subunit beta	147.5	161.5	5.6	0.0
Cluster-7222.7	*GSTUP*	Glutathione S-transferase U25	333.6	304.3	4.4	0.0
Cluster-7227.1	*RS202*	40S ribosomal protein	442.1	546.1	6.7	0.0
Cluster-7844.0	*ZCF37*	ZCF37 AT1G10220; IMPGSAL1N27970	56.3	70.4	3.2	0.0
Cluster-11481.5	*G6PD*	Glucose-6-phosphate 1-dehydrogenase	156.0	470.6	3.9	0.0
Cluster-13110.1	*SD18*	Receptor-like serine/threonine-protein kinase	73.9	94.1	13.7	0.0
Cluster-23641.1	*XTH23*	Xyloglucan endotransglucosylase/hydrolase protein 23	460.3	981.2	3.8	0.0
Cluster-25384.10	*IPYR4*	Soluble inorganic pyrophosphatase 4	458.4	438.7	3.4	0.0
Cluster-26866.5	*ALA9*	Phospholipid-transporting ATPase 9	72.1	32.5	6.7	0.0
Cluster-28356.0	*CH62*	Chaperonin CPN60-2, mitochondrial	205.4	149.5	3.7	0.0
Cluster-28583.0	*PMTE*	Methyltransferase PMT14	65.8	104.8	3.3	0.0
Cluster-29004.0	*RS11*	40S ribosomal protein S11	246.9	542.3	9.9	0.0
Cluster-29660.0	*COPD*	Coatomer subunit delta	110.9	159.9	3.0	0.0
***P. vulgaris* → *T. chinense***						
Cluster-73329.0	*EFTU*	Elongation factor Tu, plastid	0.1	14.6	4.3	4.9
Cluster-83509.0	*RL401*	Ubiquitin-60S ribosomal protein	0.3	4.1	5.7	17.0
Cluster-86098.6	*BiP*	Luminal-binding protein	0.1	3.5	9.0	14.4
Cluster-43539.64	*--*	--	0.3	3.5	5.9	23.1
Cluster-92949.1	*PAT*	Glutamate/aspartate-prephenate aminotransferase	0.0	10.7	96.7	19.8
Cluster-26621.39	*YCF68*	Uncharacterized protein	0.7	65.8	18.8	3.4

TH: *T. chinense*. THC: *T. chinense* chimera. PC: *P. vulgaris* chimera. P: *P. vulgaris*.

**Table 4 plants-13-00804-t004:** The genes related to haustoria formation.

Unigene ID	Gene	Annotation	FPKM			
			TH	THC	PC	*p*
Cluster-28098.16	*1433D_SOYBN*	14-3-3 protein	277.4	291.9	0.2	0.0
Cluster-30463.1	*ACT11_ARATH*	Actin	0.3	5.2	1.7	0.0
Cluster-30642.1	*ACT1_ORYSI*	Actin	0.3	1.2	7.2	0.1
Cluster-26673.1	*AMT11_SOLLC*	Ammonium Transporter Family	234.2	428.8	2.1	0.0
Cluster-37206.1	*CAF1K_ARATH*	CAF1 family ribonuclease	294.3	412.9	0.2	0.0
Cluster-28144.0	*CAES_ARATH*	Carbohydrate esterase, sialic acid-specific acetylesterase	34.6	179.0	1.2	0.0
Cluster-40689.0	*CHIT_PERAE*	Chitinase class I	82.1	604.1	2.2	0.0
Cluster-2905.1	*ALPL_ARATH*	DDE superfamily endonuclease	243.2	350.0	2.6	0.0
Cluster-33686.3	*--*	Dehydrin	3321	6625	4.0	0.1
Cluster-25215.6	*EP1L4_ARATH*	D-mannose binding lectin	499.1	735.5	1.1	0.0
Cluster-28520.1	*DUF4228*	Domain of unknown function	309.4	405.8	0.3	0.0
Cluster-31749.0	*DUF4723*	Domain of unknown function	50.4	604.3	2.0	0.1
Cluster-28199.1	*ESSS*	ESSS subunit of NADH:ubiquinone oxidoreductase	475.7	553.4	0.6	0.0
Cluster-86868.1	*ERM*	Ezrin/radixin/moesin family	0.0	0.1	0.1	0.0
Cluster-28808.1	*BBE21_ARATH*	FAD binding domain	80.6	336.6	1.5	0.0
Cluster-31390.0	*CASL1_CANSA*	FAD binding domain	88.5	126.5	1.7	0.0
Cluster-24876.1	*DUF716*	Family of unknown function	15.3	18.8	0.7	0.0
Cluster-26009.1	*FB119_ARATH*	F-box-like	34.9	118.7	0.2	0.0
Cluster-20986.5	*GSTF_HYOMU*	Glutathione S-transferase, C-terminal domain	249.2	502.4	1.3	0.0
Cluster-20103.7	*GADPH*	Glyceraldehyde 3-phosphate dehydrogenase	395	576.7	0.8	0.0
Cluster-23641.1	*XTH23_ARATH*	Glycosyl hydrolases family 16	460.3	981.2	3.8	0.0
Cluster-26468.1	*ERLL1_ARATH*	Hydrophobic seed protein	136.7	490.7	0.1	0.0
Cluster-29998.1	*LEA14_GOSHI*	Late embryogenesis abundant protein	425.5	842.2	0.2	0.0
Cluster-28203.4	*GILP_ARATH*	LITAF-like zinc ribbon domain	67.1	80.9	0.1	0.0
Cluster-27980.8	*FPPS1_LUPAL*	Polyprenyl synthetase	43.3	217.8	0.2	0.0
Cluster-29223.2	*MSK3_MEDSA*	Protein kinase domain	204.5	265.7	1.2	0.0
Cluster-26011.1	*CRK7_ARATH*	Protein tyrosine kinase	37.4	46.4	0.5	0.0
Cluster-15047.1	*SPE1_PEA*	Pyridoxal-dependent decarboxylase	489.5	618.4	1.0	0.0
Cluster-25493.2	*RL72_ARATH*	Ribosomal L30 N-terminal domain	323.6	331.5	0.3	0.0
Cluster-28869.1	*RL72_ARATH*	Ribosomal L30 N-terminal domain	139.9	210.5	0.1	0.0
Cluster-27350.3	*RL3_ORYSJ*	Ribosomal protein L3	1144	1163	0.7	0.0
Cluster-25988.1	*RL262_ARATH*	Ribosomal proteins L26	382.7	481.3	0.3	0.0
Cluster-29252.2	*RICI_RICCO*	Ricin-type beta-trefoil lectin domain	765.1	2370	1.4	0.0
Cluster-29448.5	*CSE_ARATH*	Serine aminopeptidase, S33	6.9	43.7	0.0	0.0
Cluster-72533.2	*STC*	Stanniocalcin family	0.1	0.5	1.0	0.9
Cluster-37697.0	*TCTP_ELAGV*	Translationally controlled tumour protein	8295	9521	5.7	0.1
Cluster-25717.1	*SSRA_ARATH*	Translocon-associated protein (TRAP) alpha	89.6	106.1	0.9	0.0
Cluster-27790.5	*LHT1_ARATH*	Transmembrane amino acid transporter protein	32.4	138.4	0.9	0.0
Cluster-30943.0	*TBA_EUGGR*	Tubulin C-terminal domain	0.1	0.5	2.2	0.0
Cluster-43188.0	*TBB_CHLIN*	Tubulin/FtsZ family, GTPase domain	0.4	1.7	4.4	0.1
Cluster-26035.2	*U73D1_ARATH*	UDP-glucoronosyl and UDP-glucosyl transferase	53.1	120.9	0.2	0.0
Cluster-44341.8	*ZFP*	Zinc finger C-x8-C-x5-C-x3-H type	0.0	0.1	0.3	0.0
Cluster-27579.1	*EXLB1_ARATH*	Expansin	294.2	1724	1.2	0.0
Cluster-92147.2	*ARI4_ARATH*	E3 ubiquitin-protein ligase	0.1	0.1	0.7	0.1

TH: *T. chinense*. THC: *T. chinense* chimera. PC: *P. vulgaris* chimera. P: *P. vulgaris*.

## Data Availability

The RNA-seq raw data used in this study have been deposited in Sequence Read Archive (SRA) database in NCBI under accession number PRJNA1054813 (https://www.ncbi.nlm.nih.gov/sra/PRJNA1054813, accessed on 31 December 2023, which will be released upon publication).

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
