# Peer review of "Exploring Information Exchange between Thesium chinense and Its Host Prunella vulgaris through Joint Transcriptomic and Metabolomic Analysis"

_plants, 2024, doi:10.3390/plants13060804_

Round 1

Reviewer 1 Report

Comments and Suggestions for Authors

This idea of study was novel and well-designed. However, there were a few critical issues that needed to be considered before publication:

  1. To claim the exchange of metabolites or transcripts, the authors should consider sampling at infection sites and no-infection sites; for example, the authors should split the root into the left and right sides and consider the primary root as the middle point. If the parasitism occurred on the left side, the right root samples would be used for comparisons. The way used in the manuscript was too uncertain to conclude the exchanges. They should at least perform qRT-PCR to check the genes.
  2. Although four groups were distributed well in the PCA and heatmap, no pattern showed metabolite exchanges after infection. The authors should discuss this in the text.
  3. The author should discuss the limitations of this work and propose clearly for the future direction.
  4. The English in the text should be a minor check.
Comments on the Quality of English Language

Minor English editing is required. 

Author Response

1Answer: Thank you for your professional review of the manuscript. For this paper, the purpose of our study is to explore the underlying mechanisms of information exchange between Thesium chinense and its host Prunella vulgaris. By employing independent Thesium chinense and Prunella vulgaris as control groups, we analyzed the changes in metabolites and genes of both species following the establishment of their parasitic relationship. Moving forward, we plan to expand our investigation by conducting research at various sampling sites, including infection and non-infection sites, to gain a more comprehensive understanding of the information exchange dynamics between Thesium chinense and its host.

Regarding that the expected time of revision is Mar8 and the object of this manuscript is identifying potential mobile mRNAs and haustoria formation-related genes through bioinformatic analysis, double-checking the genes by experimental qRT-PCR is impossible. As an alternative, we have cross-checked the results of mobile mRNAs and haustoria formation-related genes using BLAST and FPKM.

2Answer: Thanks for your suggestion. The metabolite exchange was explained from the aspect of ‘transferred metabolites’which were undetected in TH or P but were observed in other three samples, while the heatmap displayed the differentially accumulated metabolites (DAMs) in T. chinense and its host P. vulgaris post infection.

3Answer: Thanks for your suggestions. We have included the limitations of this work and proposed future research direction in the last paragraph of Discussion section, which we hope will be acknowledged.

4Answer: We appreciate for your warm work earnestly. We tried our best to improve the manuscript and made changes to the manuscript, especially scientific writing.

Reviewer 2 Report

Comments and Suggestions for Authors

Study is sufficiently organized and methods are appropriate. General English usage and grammar should be improved throughout so that the merit of the work can be appreciated. 

Comments on the Quality of English Language

The writing makes the work more difficult to engage with and appreciate. 

Author Response

Thanks for your suggestions on improving the performance of our manuscript. Sorry for the inadequate description of Materials and Methods. We have revised the Methods section (subsection 4.6 and 4.7) according to Editor and your suggestion on “incorporating comprehensive details on bioinformatic procedures and databases”, and the corresponding Results were also revised.

Round 2

Reviewer 1 Report

Comments and Suggestions for Authors

Thanks for the efforts to answer my questions.